# Do Smart Cities Represent the Key to Urban Resilience? Rethinking Urban Resilience

**DOI:** 10.3390/ijerph192215410

**Published:** 2022-11-21

**Authors:** Simona Andreea Apostu, Valentina Vasile, Razvan Vasile, Joanna Rosak-Szyrocka

**Affiliations:** 1Department of Statistics and Econometrics, Faculty of Statistics, Cybernetics and Economic Informatics, Bucharest University of Economic Studies, 010552 Bucharest, Romania; 2Institute of National Economy-Romanian Academy, Calea 13 Septembrie 13, 050711 Bucharest, Romania; 3‘Costin C. Kiritescu’ National Institute of Economic Research, Calea 13 Septembrie 13, 050711 Bucharest, Romania; 4Department of Production Engineering and Safety, Faculty of Management, Czestochowa University of Technology, 42-201 Częstochowa, Poland

**Keywords:** resilience, smart city, urbanization, correlations, regression analysis, principal components analysis, cluster analysis, Europe

## Abstract

The pandemic disrupted all activities, so it became necessary to understand, but also rethink, the complexity of economic resilience to better deal with future shocks. A component that can signal the resilience potential of a socio-economic system is smart city response, using technology to make services more efficient. This paper aims to analyze the relationship between smart cities and urban resilience to determine whether urban resilience is significantly influenced by urban smartness. Given the EU SDGs Strategy and the Implementation of RRF Programs, we have also identified the main driving forces that can amplify the impact of smart city development policies on local resilience. The results highlighted that at the European level, smart cities are significantly correlated with urban resilience; urban resilience is explained by the variation in urban smartness; resilience is correlated with all smart city dimensions, highly in (post-)pandemic, redefining a “new normal” in resilience approaches for smart cities. We also stressed the emerging, more complex content of the economic resilience concept and the new structural approach of smart cities resilience for the post-COVID-19 period.

## 1. Introduction

The complex impact of the pandemic and how this extreme event has influenced urban sustainability are still to be understood. At present, over half of the world’s total population lives in urban areas and this will increase by over 2/3 by 2050 [1]. For over a year, there were dramatic changes worldwide in terms of lifestyle and use of digital devices, i.e., working from home, changing the urban landscape, with a lasting impact on transport/other daily services. Economic recovery after shocks is not good enough and previous crises have demonstrated the fragility of the economy and the unsustainable effects on social and societal components of development. So, it has become necessary to draw up a robust recovery plan to enable resilience at the city level, especially in terms of the support provided by cities during the pandemic, for citizens.

Thus, issues such as mobility, accessibility, the increased importance of green spaces, and the role of locally sourced food are targeted. Resilient cities are cities that able to absorb, recover and prepare for future shocks (economic, environmental, social and institutional), as well as favoring sustainable development, well-being and inclusive growth [2].

Starting with the factors associated with current smart city components, it became obvious that is necessary to redefine “community recovery”. Thus, our research aims at: (a) analyse the resilience of smart cities in terms of effects of the pandemic work and life in cities, identifying the extent to which changes are needed and areas of activity that have practically been redefined; (b) determine the connection between urban resilience and urban smartness and identify whether smart cities are the main drivers in achieving community resilience; (c) an analysis of the extent to which smart cities resilience is supported by funding through RRF Programs, given the gap in Member States’ development and their own capacity to build resilience and robust recovery mechanisms; (d) an analysis of resilience taking into account the relevance of health components after the first year of the pandemic and the challenges of the post-pandemic period based on statistical indicators at the European level.

The results will allow us to define whether the direction and content of the measures are addressed and contribute to the development of a new (higher-resilience) model of smart cities or whether they only support a post-crisis fragile quick recovery. Extreme events, such as pandemics and inconsistent/unrelated economic, social and societal developments, have become the current realities of community life and should therefore be managed through: (a) prevention and intervention to improve the immediate effects and (b) good management of the indirect/adverse effects, in terms of the medium/long term, i.e., reducing inequality, correlating economic efficiency with social inclusion and social responsibility. These are considered by the authors as key factors for future development, supported by investments in technology, skills, youth employability and community based policies.

Therefore, this paper is structured as follows. The literature review presents an overview of select studies regarding smart cities and urban resilience, while Section 2 is dedicated to exploring the European national PRR content and Romanian national PRR content using a bibliometric analysis. Section 3 presents the methods used in the analysis. Section 4 presents information related to the data used in the analysis and the main empirical results. Discussions and main conclusions are summarized in the Section 5.

The findings highlight the relationship between smart cities and urban resilience as well as the need to converge towards a “new normal” that is more resilient, outlining the importance of health components in longlasting urban qualitative development.

### 1.1. Literature Review

Given the paradigm shift in resilience in the pandemic context, in order to achieve urban resilience, smart cities are necessary.

Even if there is no agreed upon definition of resilience [3], there is consensus that the concept includes the diffusion of ICT as well as “people and community needs”. Moreover, resilience differs from other limiting concepts, such as a “digital city” [4] or an “intelligent city” [5,6], having a more integrative approach, including old and new economic, social and societal components. The components of the smart city concept are develop according to digitalization dynamics and permanent market diversification with technological products/services. There is also a second level to the approach which associates the components of the smart city concept with available local resources and the level of development; therefore, smart city resilience should be tailored to a particular vision, priorities, and barriers.

Discussions in the literature regarding the concepts of urban resilience, a resilient city, a smart city, and smartness are current, still ongoing, with nuances and interpretations. Tzioutziou and Xenidis [7] present a recent synthesis of the literature, seeking to determine “the potential of the conceptual convergence between resilience and smart city frameworks”. The authors highlight common components and specificities, accept the emerging nature of concepts and measuring instruments, and conclude that, on the one hand, the urban resilience and smart city concepts “serve the goal of sustainability and share the operational framework of systems’ thinking” and, on the other hand, “the systemic capacities of adaptation, efficiency and knowledge creation are common in the frameworks of urban resilience and smartness”. Therefore, this paper contributes to the current debates and adds to the emerging elements/components, such as innovation and adaptation. In this paper, we emphasize the need to approach the resilient development of a smart city from the perspective of the robustness of the development directions and the integration of aspects of flexibility and smartness. Starting from the current definition of the term given by the Cambridge English Dictionary associated with the future development of cities, we consider a smart city as one with the ability to react quickly and in an innovative/adaptive way to overcome difficulties, through a robust approach, based on a flexible, sound community-based policy.

#### 1.1.1. Smart Cities Resilience—An Emerging Concept

Fundamentally, resilience refers to ongoing adaptation in order to recover the necessary balance and performance, “despite experiencing adversity” [8]. Resilience has emerged as an attractive perspective regarding cities, often theorized as highly complex, adaptive systems [9,10]. To better prepare us to deal with future shocks, it is necessary understand how resilience can be employed. To date, resilience was considered as a more theoretical and abstract framework guiding risk management, but nowadays we should focus on resilience planning and practice. Therefore, we need to rethink resilience in order to enhance adaptive governance, a learning-based and flexible decision-making process.

Economic resilience today means more than resilience from the classic approach, which is also true of mechanical resilience (hence we initially adopted the principle). As the economic recovery after the 2008 crisis proved to be fragile, with many components being unsustainable, there is currently a growing consensus on the need for robust, resilient and sustainable economic recovery. The post-2008 crisis evolution and the post-COVID-19 economic relaunch efforts so far have highlighted that recovery after any crisis follows a new pattern, which incorporates other partial balances using macro tracking indicators. Moreover, structural changes are insufficient and generate adverse effects if they are not correlated and if the new development model has not created its own, internal, self-sustaining mechanisms, i.e., continuous domestic demand, performance-based consumption, investment in future needs and consistent improvement in the working and living conditions of all. The reality of the last half century has shown that the sum of development components does not ensure corresponding development of the total; on the contrary, the lack of complementarity in strategies and policies generates adverse effects, i.e., limiting national potential human growth, increasing inequality [11,12,13], inefficiency in facing crises [14,15,16], and extreme behaviors in promoting cultural identity (social problems in integrating ethnic communities, return to social practices that violate human rights, and political excesses) [17].

Both the 2008 crisis, which was generated by an internal factor—the financial sector—and the COVID-19 crisis caused by an external factor—the pandemic—have disrupted the fragile equilibrium of society and highlighted the weaknesses of the socio-economic system. Both crises demonstrate the complexity and intensity of the impacts. Hence, robust and resilient recovery is being discussed with the aim of avoiding the past mistakes of immediate growth through consumption and palliative measures for a series of economic and social imbalances. So, the economic recovery initiatives follow, as the literature has already mentioned, "a new present", identifying a new recovery algorithm for each component affected by the crisis. The concern for resilience in the community is not new, there are examples from the last decades of the previous century [18,19]. Resilience for community well-being includes not only the targets, but also “requires attention to processes, as well as to outcomes, … integrates economic and social goals, and fosters connections across diverse groups within its borders” [20]. Resilience has become very popular lately [21], generating information about socio-ecological systems and their sustainable management [22], especially on climate change [23,24,25].

Smart cities resilience, in the modern approach of economic resilience, imply an achievement and overcoming of economic, social, and cultural performances as a new developmental balance, based on the association of factors and resources on another growth pattern, capitalizing past experiences—good and bad—and rebuilding subsystems, such as creative disruption, which means robust and sustainable recovery and smart development. It is in fact a new development paradigm, in which economic resilience must be understood correctly, completely integrated, as an objective of human development and not only as mechanical resilience.

The resilience of cities with an integrated vision that meets the requirements of the future involves:Current economic-social and societal development, assumed by a community roadmap for local sustainable development and sound policies, promoting investment initiatives and integrated hubs for local growth, from a strategic perspective.Supporting the development pillars that capitalize the specificity, authenticity and self-sustaining potential of development: industries, including cultural heritage [26], insufficiently exploit natural and anthropic potential, and valorize existing human capital and demographic support, both by increasing the native population and the mobility of people (internal mobility or immigrants) [27].Promoting FDI to complete/diversify economic and social activities and to retain and develop local skilled labor.Local good governing must include at least three components: digitalization and intelligent development of community services, transport and communication, and quality of life.The health network under the responsibility of the local community is to focus on services in order to increase quality of life in the metropolitan area and areas of regional influence, specializing on the risk of zonal medical diseases, in order to promote the public–private partnership in preventive, curative and recovery health services, so as to be complementary to the network of health services of national/international interest.The education network is to support the educational excellence in specialization fields required by the local and zonal labor market in order to retain the young generation through attractive jobs as alternatives to external migration—whether it is temporary for work or definitely. University education and continuing education services should ensure integration into the national education network as a center of excellence in training or specialization. The education sector should promote the connection between school and businesses through scholarships, internships and pre-employment, increasing the efficiency of the educational act and good management of the structural demand of the labor market.

A smart city appeared as a solution to urbanization and urban development. Urban desolation implies the development of the city from economic and social perspective, a reason for attracting people, thus appearing urbanized. As with any phenomenon, urbanization also has disadvantages, affecting the environment, traffic, governance, living conditions, economy and people. Thus, a smart city can be seen as a whole that solves all these problems, a synergistic interconnection of the emerging components [28].

As the smart city concept evolved, an implementation model was developed, and the characteristics of a smart city are presented in Table 1.

#### 1.1.2. Urban Resilience vs. Smart City Resilience—Debate and Main Developments in the Pre-Pandemic Period

Urban resilience reflects the ability of an urban system (including socio-ecological and socio-technical networks) to maintain or quickly return to initial functions, to rapidly transform systems for current or future adaptation [21]. The basis of urban resilience is four pillars: recovery, adaptation, transformation and resilience [31]. Urban resilience can be defined differently, depending on the perspective analyzed: (a) the engineering perspective—urban resilience is the capacity of a city to absorb change or stress and return to its previous state [32]; (b) the ecological perspective—urban resilience is the capacity of a city to adjust to a shock or disaster without damaging the existing structures and relationships [22,32,33,34]; (c) the socio-ecological (evolutionary) perspective—urban resilience is the capacity of a city to adapt or transform in response to a change or shock [35,36,37]. De Falco et al. [38] highlighted that a smart city and a resilient city are based on similar development trajectory systems, but are connected [39,40]. There is also evidence that some smart cities are not much more resistant to unexpected events such as climate change or disasters. Zhu et al. [41] analyzed 187 smart cities in China, concluding that the overall resilience was relatively low.

The transformation of traditional cities into smart cities starts with digitalization, related emerging technological frameworks, Internet of Things (IoT) and big data (BD). The outcomes are better public services for people and better use of resources, with minor impact on the environment. The following are several formal definitions for the smart city we mention: (a) a city connecting different infrastructures—physical, ICTs, social, and business—in order to capitalize the “collective intelligence of the city”; (b) “a sustainable smart city” is an innovative and digital city that can “improve the quality of life, urban operations and services and the competitiveness, while ensuring the needs of present and future generations on economic, social and environmental issues" [42]. Winters [43] analyzed the development factors that influence smart cities, from the migration perspective. Most smart cities are often higher education centers that grow due to migrants who often remain in the city after completing their studies. Additionally, the demographic growth of smart cities is based on national-level population mobility, but the overall effect is modest.

Caragliu et al. [44] realized a geographic map of smart cities for the EU27 and analyzed the factors that determine their performances. The results indicated that urban wealth is significantly influenced by the presence of a creative inputs, urban environment quality, level of education and accessibility and use of ICTs for public administration.

Ismagilova et al. [45] investigated smart cities from a systems information perspective, analyzing smart mobility, living, environment, citizens, governance and architecture, and related technologies and concepts, with the results indicating smart city alignment with ONU SDGs. Recently, Barns [46] addressed the role of urban data platforms as key sites for developing new governance models for smart cities and forums, where decision makers, researchers, urban planners, and technologies are trying to test the potential and pitfalls of data-driven methodologies in addressing contemporary urban challenges.

Zanella et al. [47] conducted a comprehensive survey on technologies, protocols and architecture that activate an urban IoT in the Padova Smart City project. The results underlined that IoT in a smart city resolves the challenges generated by the exponential growth of urbanization and population, intelligently supporting city operations with minimal human interaction [48].

Lopes [49] conducted six interviews with people involved in Smart Cities projects in Brazil, Singapore, Colombia, Portugal and Uruguay in order to analyze governance models implemented in smart cities. The results empirically highlighted that smart cities and e-government have a similar trend, with both converging to intelligent governance. Declining population, infrastructures and budgets affect cities, with the solution being using new technologies, promotion of innovation and knowledge management—a smart city [50].

The process of creating and capitalizing knowledge involves three factors: university, industry and government. Lombardi et al. [51] analyzed the link between these factors, and the influence on smart city performance. As methods, they used modeling, clustering and an analytical network process. The model obtained allowed interactions and feedback within and between clusters, with the result being a report that scales the priorities from the elements.

Creating a "smart" city is a strategy in order to mitigate the problems generated by the growth of the urban population and rapid urbanization. Chourabi [52] identified eight critical factors of smart city initiatives: management and organization, technology, governance, political context, people and communities, economy, infrastructure, and natural environment. These factors support an integrative framework that can be used to examine the initiatives of local governments regarding smart cities. The framework suggests directions and agendas for smart city research and has practical implications for government professionals [52].

One major disadvantage for urban development is pollution, causing many dangerous effects on human beings. The difference between a developed city and a smart city is the that a smart city protects the environment. Muhammad et al. [53] implemented WSN (Wireless Sensor Network) nodes for constant monitoring of city air pollution and the movements of buses and public transport machines. Data were analyzed when buses and cars reached the source destination after passing through stationary nodes in the city. Innovative mesh architecture is a more efficient way of collecting data from WSN nodes, and has many benefits in terms of smart cities.

A mobile multiplier-based environmental monitoring system, developed in Wuhan to better monitor the urban environment is a supplement to fixed environmental monitoring stations. The mobile sensors generate measurable digital images, thus making quick measurements from a distance to locate and determine the dimensions of objects and to facilitate presenting and analyzing the environmental data [54].

Vehicle transport is also an important part of modern cities. Developed cities are facing many road accidents and congested traffic, which represent obstacles in order to achieve smart cities. Ning et al. [55] showed that vehicle networks (VSNs) are capable of solving these problems, enabling intelligent mobility in modern cities. Benevolo et al. [56] analyzed the role of ICT in supporting the actions of intelligent mobility, and how quality of life was influenced. The culture of smart people in a smart city deliberately decides that they also act in order to create a smart economy [57].

Summarizing, Figure 1 highlights the most relevant studies focusing on smart city components and Figure 2 presents an overview of the most relevant studies regarding smart cities and resilient cities.

So, transforming a city into a smart city implies a smart and sustainable living space, socially, economically and environmentally. A smart city offers a friendly environment, through public services, technical and social infrastructure, a high level of security and care for the environment and green areas [58]. Urbanization has different characteristics on each continent, requiring different approaches, policies and strategies. Each city in a specific country may have different challenges for the economic development of smart cities [57].

#### 1.1.3. Measuring Cities’ Resilience the after Pandemic Shock

Pandemics are community shocks that affect both the short-term and medium- and long-term work and life patterns. In the short term, economic, social, behavioral restrictions are aimed to reduce and eliminate its effects. In the medium and long term, pandemics are extreme events (as are natural events such as earthquakes, or extreme events generated by human society such as conflicts), but which can be avoided or delayed or/and the impact can be diminished (through prevention actions), but for this we must change the life model, adapting by incorporating (a) changes that ensure prevention; (b) a reduction in negative impacts in the short term by technical unemployment, online education for citizens, limiting traffic in the community; and (c) managing the side effects that usually become evident and spread later such as increasing inequality and vulnerability of citizens, changing the business model, and fortuitous reform in some sectors, e.g., health, education, and transportation).

In fact, we are witnessing a change of approach in the strategy of sustainable development of localities, with robust and resilient recovery implying a new model of community development, a new vision, based on technology, skills, flexibility, and endogenous development.

The response of cities after the pandemic shock already offers lessons for future resilient development. The smart cities’ development process was not stopped with the pandemic; on the contrary, the pandemic was the pivotal factor of urgency for the reforms, not only by immediate adaptation to various restrictions, including lockdown, but also to “re-orient innovation teams against the virus as a way to improve government while simultaneously addressing the most dangerous emerging threat to the community” [59]. Starting with the health and educational components, step by step the “quick respond measures” included all essential services. The first step was to identify a shared vision and commitment in order to solve the pandemic crises, first aiming going back to “business as usual”. However, smart city reboot meant the emergence of "a new normal”, especially as we are currently facing wave 4 of the pandemic and the rapid diversification of SARS-Cov-2 variants (from alpha to lambda variants, etc.it is still a process in evolution).

Digital transformation was the main concern for solving problems, starting from redesigning the urban business model and dealing with surviving budget shortfalls and social policy reframing. The immediate effects of the pandemic lockdown, increasing inequality and increasing unemployment, led to the need for a radical re-think of the city, i.e., “15-Minute City–based on sustainability, resilience and place identity” [60] or lasting sustainable recovery based on “More Just, Green and Healthy Future”, respectively, focused on “4 key priorities: Rethinking the Form and Function of the City; Addressing Systemic Poverty and Inequality in Cities; Rebuilding a ‘New Normal’ Urban Economy; and Clarifying Urban Legislation and Governance Arrangements” (UN-HABITAT, 2021) in a wider perspective of enforcing regulatory capacity to prevent and react (EP, 2020).

The pandemic showed us that:-Cities were unequally developed from the perspective of modernization and transition to smart cities and hence there were differentiated efforts for digitization;-The level of economic development has substantially influenced the efficiency of health sector adaptation response to the pandemic pressures;-The management of inequality and social policies was, in fact, a residual concern, hence the inefficiency of the government in managing major social problems in the community to overcome the effects of the pandemic;-Cities that adapted best were those that were initially functional and able to deliver high-quality public services for all people [61], as well as those that efficiently governed at the beginning of the pandemic; further, those that could respond the best to the challenges and risks during the pandemic were those who managed to redistribute their funds, i.e., an adequate prioritization of interventions on most affected activities, e.g., smart financing.

In order to analyze the resilience of the smart city after the pandemic, we proposed a new normal regarding smart cities that takes into account the incidence of the pandemic, including health as an independent component. Thus, “new smart recovery” can be the result of: (1) health for all, (2) economic recovery, (3) social resilience, (4) quality of life in the community, and (5) efficient government.

Based on previous theoretical considerations, in order to identify if resilience is correlated to smartness, if smart cities are conducive to urban resilience, and if health should be considered a core component of smart cities in order to be conducive to better resilience, the following hypotheses (H) have been created to stipulate how smart cities lead to increased resilience in urban areas:

**H1:** 
*There is a strong link between smart cities and resilient cities in the case of European countries.*


**H2:** 
*Smart cities are conducive to more resilient cities in the case of European countries.*


**H3:** 
*Smart city components are strongly correlated with urban resilience.*


**H4:** 
*The “new normal” regarding smart cities, considering the five components (including the health as separate one), is conducive to better resilience.*


## 2. Data and Methodology

In order to determine the relationship between smart cities and resilient cities, we analyzed two global indices for European countries for 2020 and 2021 on resilience and smartness, with this paper being the first attempt regarding smart cities and urban resilience.

In order to characterize urban resilience, we used the Urban Resilience Index (URI), best reflecting the level of resilience. It is a composite measure calculated at the country level (it takes values from 0—the lowest resistance to 100—the highest resistance) and includes three basic resilience factors: economic, risk quality and supply chain. Additionally, each factor represents the result of four basic factors.

The index that best characterizes the smartness regarding cities is the Smart City Index (SCI). It reflects residents’ perceptions of issues related to the technological structures and applications available in their city. It includes 109 cities around the world and the perceptions of 120 residents in each city, and the final score for each city is calculated using the last two years of the survey. Residents’ perceptions focus on 2 pillars: existing city infrastructure and technology, which describe the technological supplies and services available to residents, and each pillar is assessed on five key areas: health and safety, mobility, activities, opportunities and governance. This represents a ranking, taking values from 1 (meaning the smartest city) to 109 (the least smart city). This ranking approach can be used as an effective instrument detecting strengths and weaknesses and improving a city’s smartness [62].

The data are provided by the Institute for Management Development, in collaboration with Singapore University for Technology and Design (SUTD) for the SCI and FM GLOBAL in the case of the URI. The data are processed using EViews 9.5, SPSS and Tableau.

In order to analyze the most common words regarding the national RRF Program, we used a bibliometric analysis. A bibliometric analysis investigates the content in a systemic and systematic process, structuring and ordering the results obtained and converting this from qualitative to quantitative data.

Bibliometric methods provide a quantitative analysis for written publications, i.e., “infometrics” [63,64] and “scientometrics” [65]. This analysis involves the identification of the literature content, i.e., within a given subject area. Therefore, the scientific production is evaluated, with the results being of major importance to policy makers, scientists, and other stakeholders [66]. A bibliometric analysis is considered a state-of-the-art methodology, including components from all scientific domains [67].

In order to identify the main topic of the content, we used word clouds based on the frequency of words included. The relationships between words can be determined, investigating which words tend to follow others immediately, or that tend to co-occur within the same documents. Both types of analyses are complementary. If word networks reveal which word pairs co-occur most often, correlation networks reveal which words appear more often.

Performing the correlation analysis, we determined if there is a link between smart cities and resilient cities [68,69]. In order to highlight if smart cities are influencing resilient cities, we performed a regression analysis in order to measure the bond that exists between variables, and identify the relative law according to the links between variables [68]. In our case, the regression model can be written as follows:Resilience Index = β_0_ + β_1_∙SmartCityIndex (1)

In order to group the countries according to the SCI and the URI, we used a cluster analysis [70]. Hierarchical cluster analysis implies collection methods seeking to construct a hierarchically arranged sequence of partitions for some given object set, resulting in a hierarchy based on proximity measures defined for every pair of objects [71]. In order to highlight the link between smart city components and resilience and the implications of the new normal regarding smart cities on resilience, we used a Principal Components Analysis (PCA). The purpose of a PCA is to condense the information of a large set of correlated variables into a few variables, while not throwing overboard the variability present in the data set [72]. We used a total of 144 variables provided by Eurostat and Urban Audit, all selected by NUTS2 regions.

## 3. Empirical Results

### 3.1. A Bibliometric Analysis on Resilience and Smart Cities

In order to select the most relevant studies in the field, we used a bibliometric analysis, with the principal source of scientific articles selected being the academic platform Web of Science. We explored the content of 668 research articles related to smart cities and resilience. In order to highlight the structure of the scientific field, we used a content analysis, inspecting the most common words and the relationship between words.

Additionally, a network of co-occurrences, with a frequency of at least 20 times, was taken into account, with a correlation degree greater than 0.5. The analysis was performed using the Vos program.

Exploring the valuable information provided by the world clouds, we tried to respond to the following main research question: Which are the most common words found in the full scientific articles on resilience and smart cities?

The empirical analysis proved that the most common words in the full content of selected articles are: “resilience”, “city”, “approach”, “system”, ”development”, “smart city”, “technology”, “quality”, “service”, “policy”, and “planning” (Figure 3).

The most common combinations in the most relevant studies in the field are: city, community, planning, impact, adaptation, resilient city, resilience, system, solutions, network, internet, technology, smart city, development, future, sustainability, process, project, and opportunity (Figure 4).

In order to highlight the combination of words being encountered the most, the most correlated words within the selection of articles was explored using a threshold value of 0.5.

### 3.2. A Bibliometric Analysis on RRF Programs

In order to highlight the most common words at the RRF Programs level, we mapped the words considering all the EU countries, with the results highlighted in Figure 5. The words that appear most often, considering a higher frequency of occurrence of 20, are: social resilience, investment, reform, recovery, resilience plan, total allocation plan, business education, key measures, recovery, and COVID.

In order to highlight the most common words encountered at the Recovery and Resilience Facility (RRF) Programs level, we mapped the words considering EU countries, with the results highlighted in Figure 3. The words that appear most often, considering a higher frequency of occurrence of 20 are: social resilience, investment, reform, recovery, resilience plan, total allocation plan, business education, key measures, recovery, and COVID.

At the level of Romania, in the content of the national RRF Program, the words that appeared at least 20 times are: public, reform, health, growth, program, financing, county, strategy, environment, investments, public, and education (Figure 6).

The analysis was performed using the Vos program and a correlation degree greater than 0.5.

### 3.3. Results Regarding the Relationship between Resilience and Smart Cities

The first step of our analysis consists of checking normality. The normality analysis of the distributions was realized using the Shapiro–Wilk test. According to it, the results indicated that the URI (Urban Resilience Index) failed to pass the null hypothesis of normality; instead, the SCI (Smart City Index) reflects a normal distribution, as the boxplot (Figure 7) demonstrates too.

For measuring the correlation between the SCI and the URI, we applied a parametric method (Pearson’s linear correlation coefficient—PC) and a nonparametric one (Kendall rank correlation coefficient—KC). We based our analysis on the outcomes of the last test, and the result of the parametric method was used in order to confirm the nonparametric method results [73].

There is an inverse and strong link between the two indicators at the EU level (Figure 8). This is reinforced by the correlation coefficients (Pearson −0.78 and Kendall −0.68), meaning that a country with a high SCI will register a low score on resilience, and therefore a higher degree of resilience. So, if we reverse the scale, we can affirm that the two are positively correlated—smart cities are correlated with a higher degree of resilience, thus confirming H1.

In order to analyze if there is also a dependency relationship between the two indices, we performed a regression analysis, where the URI is the dependent variable, and the SCI is the independent variable. R squared is 0.6, meaning that the URI is explained by the SCI at 60%, thus the SCI significantly influences the URI, with a probability of 95%, confirming H2 (Table 2). Therefore, if the SCI increases with one unit, the URI decreases on average by 0.37 units, considering a probability of 95%.

Given that at the level of European countries, smart cities lead to stronger resilience, we wanted to achieve a clustering of countries in this regard (Figure 9), considering the Resilience Index and the Smart City Index. Thus, this resulted in three clusters:-Cluster 1: Germany, Netherland, Switzerland, Sweden, the UK, Austria, Norway, Finland and Ireland;-Cluster 2: Italy, Bulgaria, Hungary, Turkey, France, Romania, Ukraine, Russia, Greece and Slovakia;-Cluster 3: Spain, Czech Republic, Belgium and Poland.

In order to establish the link between smart city components and resilience, we considered the classical six components: smart economy, smart governance, smart mobility, smart people, smart living and smart environment.

All smart city components are strongly correlated with urban resilience in the affected countries, confirming H3 (Figure 10).

In order to analyze smart city resilience after the pandemic, we considered the connection between the five new pillars proposed for the “new normal” of smart cities and urban resilience.

The components of new smart cities are strongly correlated with urban resilience, leading to greater resilience (Figure 11). Therefore, the new form of resilience is the result of the five pillars proposed for the new smart city, thus confirming H4.

## 4. Main Findings and Results

In this paper, we analyzed the link between the level of smartness of cities and the degree of resilience in European countries, with the results indicating a direct link—resilience is the result of smartness. Therefore, a high level of smartness in cities implies a high level of resilience. Clustering the countries by these variables reflects a similar division to when clustered by level of economic development. In order to analyze the connection between urban resilience and smart city components, we considered the classical dimensions of smart cities (smart economy, smart governance, smart mobility, smart people, smart living and smart environment), with the PCA highlighting that urban resilience is correlated with all six dimensions. To analyze the connection between resilience and smartness in the pandemic context, we considered five new pillars for the “new normal” of smart cities and urban resilience (smart economy, smart governance, smart mobility, smart people, smart living and smart environment), with the PCA highlighting a stronger correlation with resilience than in the case of classical smartness components. Therefore, the health component is essential in order to reflect the level of smartness and, implicitly, the level of resilience and should be included when rethinking resilience components.

From the perspective of public policies that support smart city resilience, in the recent RRF Programs approved so far, all Member States except for Poland, Bulgaria, Netherlands, Hungary, and Sweden are included, depending on national specificities, short-term development needs and environment national/local initiatives that address investments in the following components:-Digital global–local connectivity—digital (local) infrastructure (Austria, Croatia, Cyprus, Denmark, Finland, Latvia, and Lithuania);-Promoting the transformation of vulnerable territories into smart and sustainable areas (Italy);-Digital transformation of businesses (Czech Republic, Denmark, Estonia, France, Greece, Ireland, Latvia, Lithuania, Malta, Portugal, Slovakia, Slovenia and Spain) including digital skills for employees (Slovenia) and digital infrastructure to streamline work- and education-based immigration (Finland);-Boosting future-oriented, transformative and innovative (national) research (Austria), with a focus on the green and digital transition (Belgium), strengthening the attractiveness of researcher’s careers (Croatia);-Education 2.0, financing a more inclusive and future-proof education system across communities focused on STEM competencies (Belgium and Romania); e-learning and digital teaching tools (Croatia, France, and Greece); reskilling/upskilling in digital skills (Czech Republic, Ireland, Latvia, Slovenia and Spain); e-training for employability of job seekers (Luxemburg), and digital infrastructure in schools (Slovakia);-E-governance with specific services for business and population (Belgium, Cyprus, Czech Republic, Denmark, Finland, France, Greece, Ireland, Italy, Lithuania, Luxemburg, Malta, Romania, Slovakia and Spain), interoperability of the government’s information systems (Croatia and Estonia), and next-generation cloud infrastructures and services (Germany);-E-health transition (Slovenia) and cross-border e-health services (Cyprus), increasing the resilience of health care services (Czech Republic, Malta, Portugal, and Romania), health smart infrastructure and cyber security (Germany, Italy, and Luxemburg) and addressing health workforce shortages (Estonia and Finland).

So, the policies of the coming years take into account the main components of smart city development and are transversal objectives in the digitalization of public services and the profound reform of the promotion of business models and product sales. The connectivity and security of digitalized systems will also shape the behavior of the population through their propensity for digital components, smartness that ensures resilience and sustainability, in its broadest sense—green, adaptive and future-oriented risk assessment). From this perspective, the research results are convergent with the general framework of the RRF, which confirms that smart cities and urban resilience are of interest and allow the achievement of national objectives for sustainable, robust and resilient development.

## 5. Conclusions

The coronavirus pandemic highlights the difficulty in facing an uncertain and unpredictable future. Although smart city resilience is not a new concept at this time, it must be updated considering the disruptive impact of the COVID-19 crisis for a robust recovery approach.

The bibliometric research aimed to identify the degree of specialists’ concern-through the number of articles identified on our topic of interest in WoS, as well as the frequency of detailed topics, components of resilience and smart cities, including the thematic content of the RRF Programs for EU countries and Romania. The analyses showed that the current national and local policies partially finance the components of smart cities or of resilience. The prioritization of investment areas depends on the level of development and local characteristics. According to the extant literature, there are some studies regarding a bibliometric analysis on urban planning, sustainability, resilience, and smart cities [74]; resilience and previous results of emerging research trends [7]; smart cities [75]; resilient cities [76]; and resilience of smart cities [77,78]. However, from our knowledge, this study is the first attempt at analyzing the association between smart cities and resilience through a bibliometric analysis.

The comparative analysis of the URI and the SCI highlighted the strong, interdependent link between smart cities and urban resilience. According to the analysis, at the level of European countries, there is a strong link between smart cities and urban resilience, with 70% of urban resilience being explained by the index that reflects the level of urban smartness. Finally, we analyzed the components of smart city recovery for the “new normal”, concluding that the five components proposed (including health as separate one) lead to a better post-pandemic resilience. Thus, resilience must be rethought, taking into account smart cities in the “new normal” context. In the literature, we found that urban resilience is often associated with sustainable development [48], but there are studies on smart cities and achieving urban resilience [38], building resilience in the urban periphery [79], integration of resilience and smart city concepts in urban systems [7], and redefining the smart city concept with a resilience approach [39]. Although there are studies on the relationship between the two concepts and the need to incorporate resilience in order to achieve a smart city, our study is the only one considering the health component, highlighting the improved resilience based on it. The usefulness of this approach is given by the comments and the results of the analysis that can contribute to the improvement and development of smart city management and deep restructuring of local strategies for implementing functional resilience in smart cities.

The results open the possibility for future research on a better measurement, through statistical indicators, of post-pandemic resilience in smart cities. The authors aim to develop and test a composite index that captures the new approach to development, through robust, resilient and environmentally sustainable recovery, from the perspective of increasing the quality of life of citizens in emerging smart cities, focused on reducing inequality and digital disruption management and increasing skilled youth employability in the community.

## Figures and Tables

**Figure 1 ijerph-19-15410-f001:**
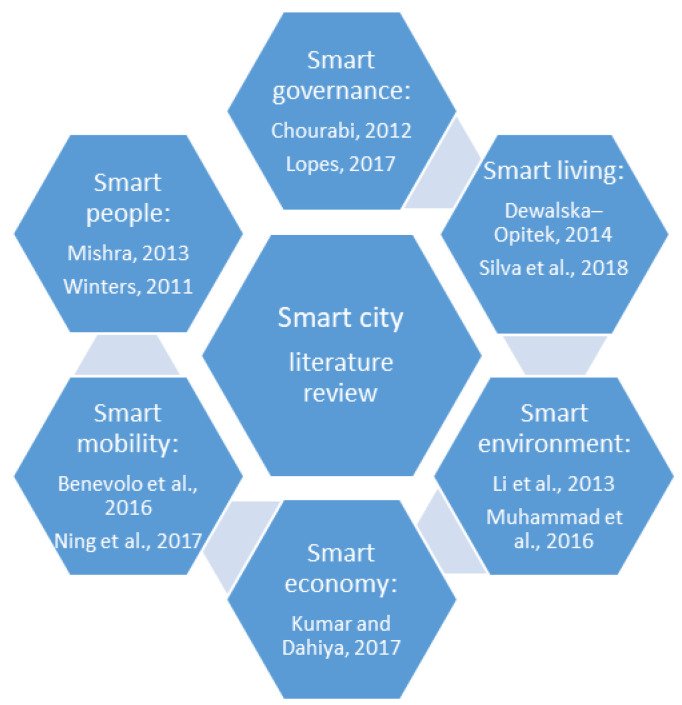
Literature review according to smart city components [43,48,49,50,52,53,54,55,56,57,58].

**Figure 2 ijerph-19-15410-f002:**
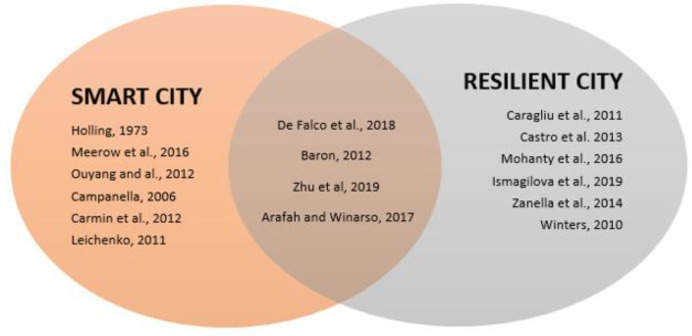
The conceptual framework of theoretical considerations between smart cities and resilient cities.

**Figure 3 ijerph-19-15410-f003:**
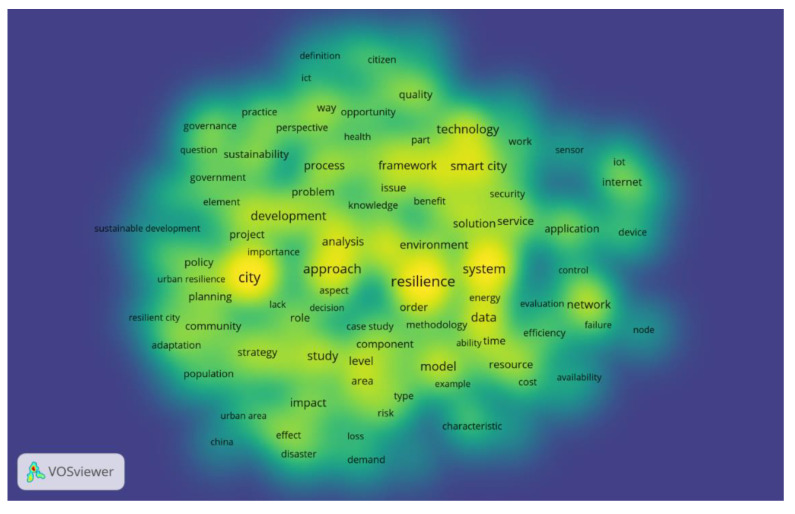
Most common words in scientific publications regarding smart cities and resilience.

**Figure 4 ijerph-19-15410-f004:**
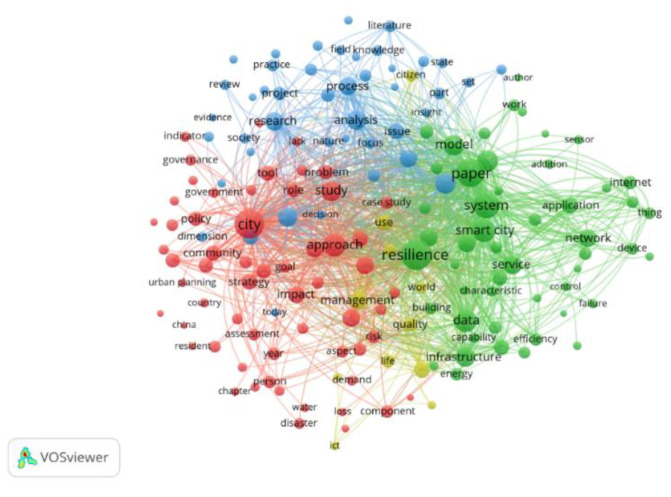
Word network in WOS publication content.

**Figure 5 ijerph-19-15410-f005:**
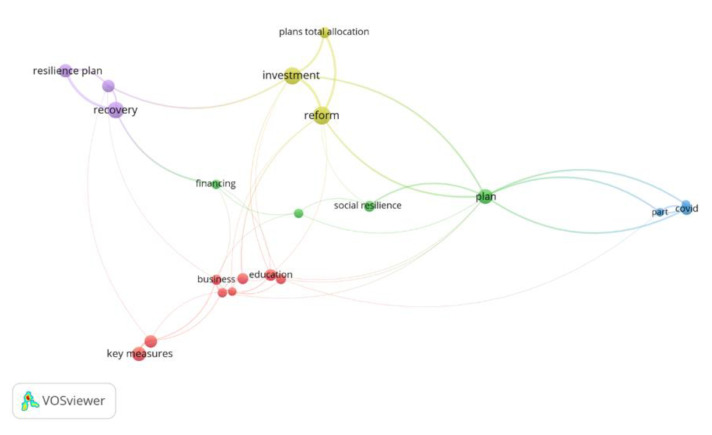
Word network in European national RRF content.

**Figure 6 ijerph-19-15410-f006:**
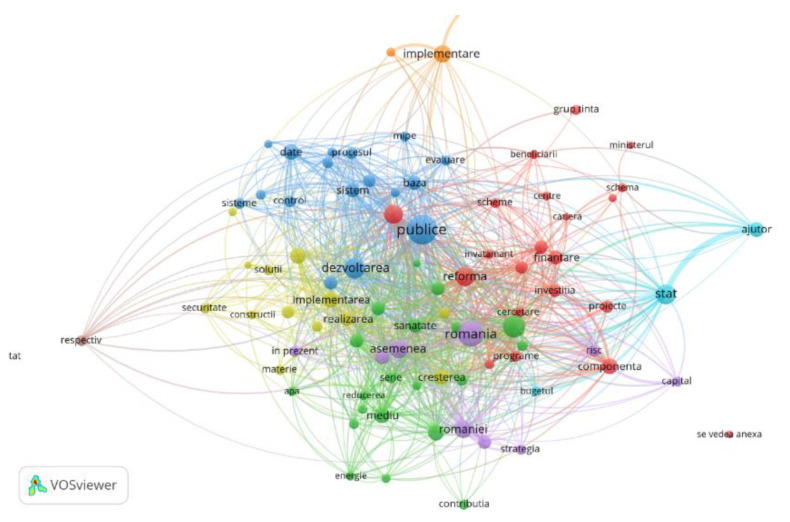
Word network in Romanian national RRF content.

**Figure 7 ijerph-19-15410-f007:**
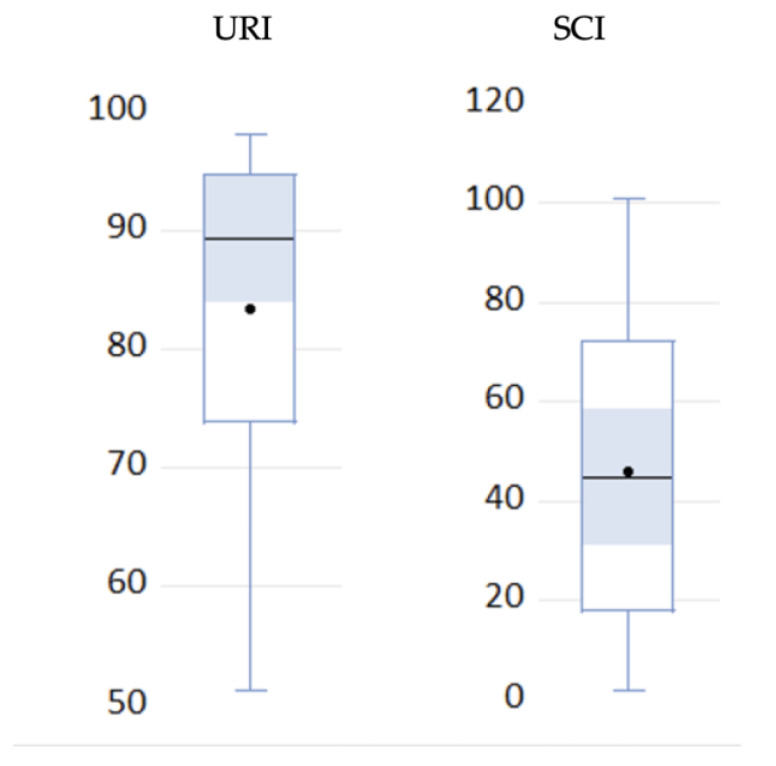
Boxplots for resilience and smart city indices.

**Figure 8 ijerph-19-15410-f008:**
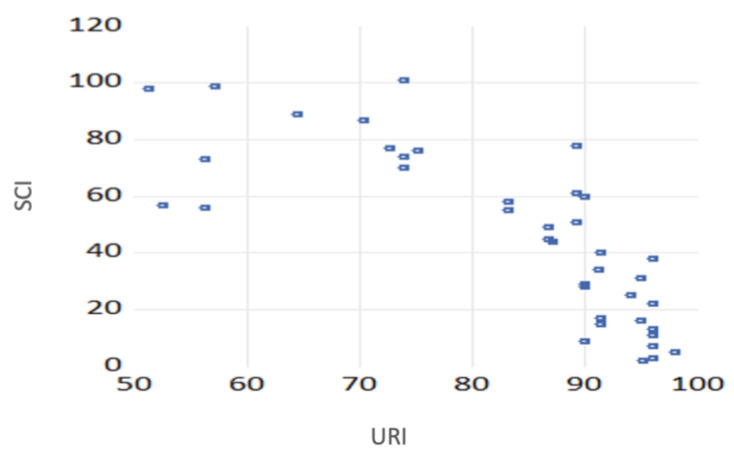
Correlogram of resilience and smart city indices.

**Figure 9 ijerph-19-15410-f009:**
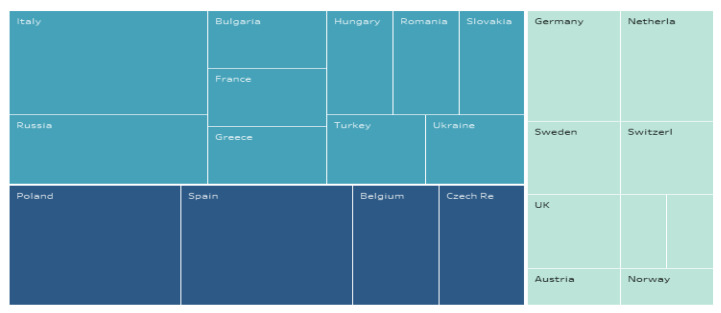
Clustering European countries according to resilience and smartness.

**Figure 10 ijerph-19-15410-f010:**
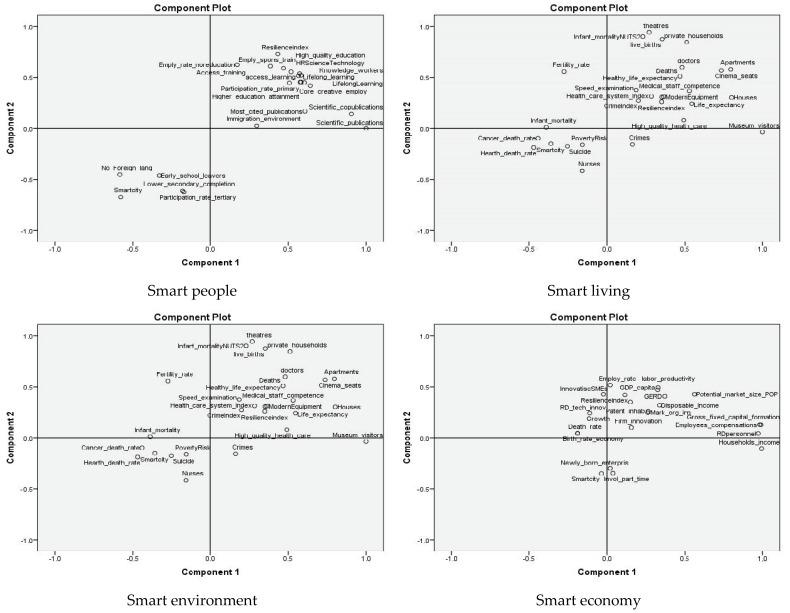
PCA of smart city components and urban resilience.

**Figure 11 ijerph-19-15410-f011:**
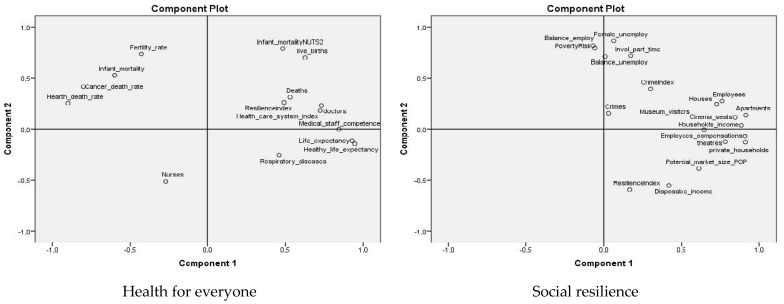
PCA of “new normal” for smart city components and urban resilience.

**Table 1 ijerph-19-15410-t001:** Smart cities characteristics—selective models.

Model	Smart City Approach	Components
2014 European smart cities 3.0 and 4.0—European medium-sized and larger cities (100 thousand inhabitants to 1 million inhabitants)	“*combination of endowments and activities of self-decisive, independent and aware citizens*” (http://www.smart-cities.eu/?cid=2&ver=3, accessed on 12 June 2022)	Smart: governance, economy, mobility, living, environment and people
Frost and Sullivan, smart cities in 2020	“*has an active presence and plan in at least five of the eight criteria*” (https://www.frost.com/wp-content/uploads/2019/01/SmartCities.pdf accessed on 12 June 2022);	Smart: governance, energy, building, infrastructure, technology, health care, citizen
World Bank’s Global Smart City Partnership Program (2021 debate)	“*is uses technology to efficiently engage citizens and meet their needs*” (https://blogs.worldbank.org/sustainablecities/5-views-what-makes-city-smart accessed on 12 June 2022)	Prioritize measures to address inequality and digital divides
OECD Programme on Smart Cities and Inclusive Growth (OECD, 2020) [2]	− A wide diversity of models based on local characteristics − Six key dimensions for smart city performance measurement	“Profitability of smart city investment and return on investment. Differences among cities in levels of economic development and on the urban value chain. Building capacity to collect and use the right data.Aligning investments with city’s strategic priorities and citizens’ needs. A multi-criteria approach.Starting small and scaling up after”
The Smart Cities Dive Outlook on 2021 [29]	“*A key lesson learned throughout 2020 is that nobody can predict the future, but that won’t stop us from trying*”. (https://www.smartcitiesdive.com/news/11-experts-predict-what-will-shape-smart-cities-in-2021/592063/, accessed on 12 June 2022)	New forced changes: − “creative transportation, including individual means for transport” − Public Wi-Fi − “tactical urbanism”, another balance between green spaces and built areas − Businesses and 5G (start-ups boom) − Digital platform for public services − E-governance
Multi-scale models for smart cities [30]	Stages of smart cities (stage 1.0 to 5.0); a roadmap for the optimized operation of smart city systems	Smart: transportation, power system, (e-)health care, community, warehouse, home, security, industry, education

Source: authors’ selection based on literature review.

**Table 2 ijerph-19-15410-t002:** Regression analysis.

	Coefficients	Standard Error	t Stat	*p*-Value	Lower 95%	Upper 95%
Intercept	100.3795	2.665042	37.66527	4.12 × 10^−31^	94.97963	105.7794
SCI	−0.36606	0.048704	−7.51589	5.89 × 10^−9^	−0.46474	−0.26737

## Data Availability

Not applicable.

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
