# Peer review of "Do Smart Cities Represent the Key to Urban Resilience? Rethinking Urban Resilience"

_ijerph, 2022, doi:10.3390/ijerph192215410_

Round 1

Reviewer 1 Report

The authors are presenting a way to correlate the Smart City Index with the Urban Resilience Index. They are starting with a sufficient (in detail) literature review and then describe the methodology that they followed and then their results. Although the description of the methodology is acceptable (section 2), the description of the results (section 3) needs significant changes.

In addition in section 2, line 439 they refer that the data used, are presented in the Supplementary Material. I could not download any supplement from the reviewer's form, please provide the supplement with the appropriate data. 

More specifically section 3 (Empirical results), needs revisions in order to be in a good level of description. Please correct the following:

1) line 458: There is no 4 section indicated, that is the number 4 here?

2) line 466: There is no 5 section indicated, that is the number 5 here?

3) Please provide in the supplement all the EU countries that are involved (line 467)

4) line 483: Please describe this paragraph better and give more details about what you did, this is not described sufficiently. For example which are these distributions that you refer to? Please provide the numbers in the supplement. 

5) Please improve figure 8 by adding axes and be of the same size of figure 7, also it will be better to replace 'smart' with SCI and 'resilience' with URI.

6) Figure 8 shows that the correlation is strong because they are a lot of points in the area of high URI and low SCI, but there are not so many points in the other direction (low resilience and high SCI). From my point of view, this creates a problem with the extract of the conclusion that the H1 criterion is confirmed. 

7) Please provide also the data of figure 8 in the supplement, are those data the same as those used in figure 7? Please also provide the names of the cities and the EU countries of these data.

8) In table 2 you provide the results of the regression analysis that you performed, I don't think that a P-value of the order of magnitude of -31 shows something, you can put it as zero. 

9) Please also provide extra information about the way you performed the cluster analysis (line 509) and how these countries are grouped. Does the color of each group show something? Please define in the text or in the legend of the figure. Also, please improve the quality of the image.

10) Figures 10 and 11 are not of high-quality (as images), the letters are very small and it is not easy for someone to follow them. Please provide also the correlation as a number for each sub-figure.

11) line 530: what is the criterion for the strong correlation (r value) in order to say that your results confirm the criteria hypothesized? 

12) line 543:  it should be section 4 and not 6. 

13) It should be section 5, not 6.1.

Other comments on the manuscript:

There is also a problem in the whole manuscript with the references. A lot of the references that are referred into the text are not included in the reference list. Please fill the reference list with all the references used. Also, the manuscript does not follow the format for the references, please check it and correct them. 

14) line 68: there is no section 4 defined

15) Please check carefully the text, there are some grammar issues.

16) in table 1, please provide the references that are missing from the reference list.

17) table 1, please use italics for the phrases which are into ""

Author Response

Dear Reviewer,

We would like to kindly thank you for your evaluation and for the constructive and copious suggestions which have helped us to improve the draft significantly.

All your comments and suggestions have been taken into account in the revised paper, as described in the following:

Comment:

The authors are presenting a way to correlate the Smart City Index with the Urban Resilience Index. They are starting with a sufficient (in detail) literature review and then describe the methodology that they followed and then their results. Although the description of the methodology is acceptable (section 2), the description of the results (section 3) needs significant changes.

Response:

We improved the empirical results section. Thank you!

Comment:

In addition in section 2, line 439 they refer that the data used, are presented in the Supplementary Material. I could not download any supplement from the reviewer's form, please provide the supplement with the appropriate data. 

Response:

We added the Supplementary Material. Thank you!

Comment:

More specifically section 3 (Empirical results), needs revisions in order to be in a good level of description. Please correct the following:

1) line 458: There is no 4 section indicated, that is the number 4 here?

Response:

We erase. Thank you!

2) line 466: There is no 5 section indicated, that is the number 5 here?

Response:

We erase. Thank you!

3) Please provide in the supplement all the EU countries that are involved (line 467)

Response:

We added another Supplementary Material with most encountered words in RRF programs for the countries in Europe. All member atates except of Poland, Bulgaria, Netherlands, Hungary, Sweden were included. Thank you!

Comment:

4) line 483: Please describe this paragraph better and give more details about what you did, this is not described sufficiently. For example which are these distributions that you refer to? Please provide the numbers in the supplement. 

Response:

We better explained this paragraph. Thank you!

countries in Europe. Thank you!

Comment:

5) Please improve figure 8 by adding axes and be of the same size of figure 7, also it will be better to replace 'smart' with SCI and 'resilience' with URI.

Response:

We modified, both Figure 7 and Figure 8. Thank you!

Comment:

6) Figure 8 shows that the correlation is strong because they are a lot of points in the area of high URI and low SCI, but there are not so many points in the other direction (low resilience and high SCI). From my point of view, this creates a problem with the extract of the conclusion that the H1 criterion is confirmed. 

Response:

The confirmation of hypothesis 1 is based also on correlation coefficients (Pearson -0.78 and Kendall -0.68).

Comment:

7) Please provide also the data of figure 8 in the supplement, are those data the same as those used in figure 7? Please also provide the names of the cities and the EU countries of these data.

Response:

The data are the same for figure 7 and Figure 8. Thank you!

Comment:

8) In table 2 you provide the results of the regression analysis that you performed, I don't think that a P-value of the order of magnitude of -31 shows something, you can put it as zero. 

Response:

We considered pvalue less than 0,05, the probability of 95%.

Comment:

9) Please also provide extra information about the way you performed the cluster analysis (line 509) and how these countries are grouped. Does the color of each group show something? Please define in the text or in the legend of the figure. Also, please improve the quality of the image.

Response:

The clusters were realized considering the resilience index and smart city index, each colour representing a cluster, being explained in cluster description.

Comment:

10) Figures 10 and 11 are not of high-quality (as images), the letters are very small and it is not easy for someone to follow them. Please provide also the correlation as a number for each sub-figure.

Response:

We modified the images. Thank you!

Comment:

11) line 530: what is the criterion for the strong correlation (r value) in order to say that your results confirm the criteria hypothesized? 

Response:

We used PCA for correlation, and the variables are very near the axis, thus the correlation is big. Thank you!

Comment:

12) line 543:  it should be section 4 and not 6. 

Response:

We modified. Thank you!

Comment:

13) It should be section 5, not 6.1.

Response:

We modified. Thank you!

Comment: 

Other comments on the manuscript:

There is also a problem in the whole manuscript with the references. A lot of the references that are referred into the text are not included in the reference list. Please fill the reference list with all the references used. Also, the manuscript does not follow the format for the references, please check it and correct them. 

Response:

We modified. Thank you!

Comment: 

14) line 68: there is no section 4 defined

Response:

We modified. Thank you!

Comment: 

15) Please check carefully the text, there are some grammar issues.

Response:

We modified. Thank you!

Comment: 

16) in table 1, please provide the references that are missing from the reference list.

Response:

We modified. Thank you!

Comment: 

17) table 1, please use italics for the phrases which are into ""

Response:

We modified. Thank you!

Reviewer 2 Report

The article presents a detailed study of the influence of a city's smartness on its resilience level. Excellent work. I would only suggest structuring chapters 2 and 3 into sections to facilitate understanding.  I would also suggest improving the quality of figure 10 as the labels are sometimes difficult to read.

Author Response

Dear Reviewer,

We would like to kindly thank you for your evaluation and for the constructive and copious suggestions which have helped us to improve the draft significantly.

All your comments and suggestions have been taken into account in the revised paper, as described in the following:

Comment:

The article presents a detailed study of the influence of a city's smartness on its resilience level. Excellent work. I would only suggest structuring chapters 2 and 3 into sections to facilitate understanding.  I would also suggest improving the quality of figure 10 as the labels are sometimes difficult to read.

Response:

We improved our paper. We added subchapters in section 3, we also improved Figure 10. Thank you!

Reviewer 3 Report

Dear author(s),

Regarding the submission "Do smart cities represent the key to urban resilience? Rethinking urban resilience", please find my comments below.

The article addresses an important topic, discussing the relationship between smart cities and urban resilience, but requires extensive revision of the text to make it suitable for publication. For instance, there are typos in the title (“reslience”), and the font (letter size and style) used in some figures is not readable. The introduction presents a sufficient background, including references relevant to the research. Moreover, the research design and description of methods sound appropriate, but the discussion of the results can be improved.

The paper is very succinct, dedicating only a few brief paragraphs, but without details, and citing few or no references to discuss the results obtained. Considering that the journal aims to reach an international audience, and also as a way of valuing the analytical arrangement and the results, it is opportune to expand the discussion to include previous studies. I recommend separating the results and discussion sections to make it more evident how the findings of the present study are related to the results of previous studies.

From the foregoing, I recommend accepting the paper after review (corrections/clarifications on these aspects).

Sincerely.

Author Response

Dear Reviewer,

We would like to kindly thank you for your evaluation and for the constructive and copious suggestions which have helped us to improve the draft significantly.

All your comments and suggestions have been taken into account in the revised paper, as described in the following:

Comment:

The article addresses an important topic, discussing the relationship between smart cities and urban resilience, but requires extensive revision of the text to make it suitable for publication. For instance, there are typos in the title (“reslience”), and the font (letter size and style) used in some figures is not readable.

Response:

We verified the entire paper and modified. Thank you!

Comment:

The introduction presents a sufficient background, including references relevant to the research. Moreover, the research design and description of methods sound appropriate, but the discussion of the results can be improved.

Response:

We improved the discussion section. Thank you!

Comment:

The paper is very succinct, dedicating only a few brief paragraphs, but without details, and citing few or no references to discuss the results obtained. Considering that the journal aims to reach an international audience, and also as a way of valuing the analytical arrangement and the results, it is opportune to expand the discussion to include previous studies. I recommend separating the results and discussion sections to make it more evident how the findings of the present study are related to the results of previous studies.

Response:

We improved the discussion section, separating from results and including in conclusions. We correlated our results with the ones in the extant literature. Thank you!

Reviewer 4 Report

I consider that the paper is correct, relevant and well written. Nevertheless, the paper is only about words incidence and semantic networks. I would like to see a real example or, at least, a discussion about how city smartness could improve resilience  concerning to a practical case.

Author Response

Dear Reviewer,

We would like to kindly thank you for your evaluation and for the constructive and copious suggestions which have helped us to improve the draft significantly.

All your comments and suggestions have been taken into account in the revised paper, as described in the following:

Comment:

I consider that the paper is correct, relevant and well written. Nevertheless, the paper is only about words incidence and semantic networks. I would like to see a real example or, at least, a discussion about how city smartness could improve resilience  concerning to a practical case.

Response:

Our results are splited in two big parts: a bibliometric analysis on the association of the words smart city and resilience and a empirical analysis in case of European countries. According to our results, in case of European countries, smart city presents a positive link with resilience, resilience conducing to smart city. Therefore, in order to achieve smart city resilience is mandatory, highlithing that resilience increases if we consider the health component. Thank you!

Round 2

Reviewer 1 Report

Comments on the revision 

1) in figure 8, you provided this time a totally different figure and I'm very confused with this because there are no numbers in the x axis. Please explain what is this, that you provided.

2) in the supplement that you provided except from the first page with the table, the other pages what are they showing? Please use legends for the tables also for the supplement. 

Author Response

Letter to Reviewer

Dear Reviewer,

We would like to kindly thank you for your evaluation and for the constructive and copious suggestions which have helped us to improve the draft significantly.

All your comments and suggestions have been taken into account in the revised paper, as described in the following:

Comment:

In figure 8, you provided this time a totally different figure and I'm very confused with this because there are no numbers in the x axis. Please explain what is this, that you provided.

Response:

We maintained the initial scale and added numbers on the scale. Thank you!

Comment:

in the supplement that you provided except from the first page with the table, the other pages what are they showing? Please use legends for the tables also for the supplement. 

Response:

We modified the Supplementary Material, adding explanation. Thank you!

Reviewer 3 Report

Dear authors,

The overall quality of the paper has improved in the issues indicated for correction. Some figures could still be improved, for instance, in Figure 2 the authors' names "Arafah and Winarso" are underlined in red. In addition, english language and style are minor spell check required.

Best regards.

Author Response

Letter to Reviewer

Dear Reviewer,

We would like to kindly thank you for your evaluation and for the constructive and copious suggestions which have helped us to improve the draft significantly.

All your comments and suggestions have been taken into account in the revised paper, as described in the following:

Comment:

The overall quality of the paper has improved in the issues indicated for correction. Some figures could still be improved, for instance, in Figure 2 the authors' names "Arafah and Winarso" are underlined in red. In addition, english language and style are minor spell check required.

Response:

We can not modify the Figure 2, this is the template. We checked the entire paper for English language and style. Thank you!